# Wearing Resistance of Metal Coating Layers after Laser Beam Heat Treatment

**Arthur Oláh [1],\* , Teodor Machedon-Pisu [1],\* and Petrică Vizureanu [2],\***

1   Materials Engineering and Welding Department, Transilvania University of Brasov, 500024 Brasov, Romania
2   Technologies and Equipment for Materials Processing Department, Gheorghe Asachi Technical University of Iasi, 700050 Iasi, Romania
\*   Correspondence: oart@unitbv.ro (A.O.); tmache@unitbv.ro (T.M.-P.); peviz2002@yahoo.com (P.V.)

**Abstract:** Laser heat treatment (LHT) is applied herein after coating. Evaluation of the results was performed by studying the microstructures via metallographic SEM/EDX microscopy, and the mechanical properties were analyzed in terms of microscopic hardness and abrasion resistance. The objective of this study was to investigate the effect of LHT on the wear resistance of metal coatings. The results indicate the influence of the microstructure and chemical composition of the electrodes on the microhardness and wear resistance of the metal coatings (MCs).

**Keywords:** metal coating; laser heat treatment; microhardness; wearing resistance





## 1. Introduction

The MC process protects metal parts and structures from damage caused by operating conditions. Metals such as steel, stainless steel, aluminum, and titanium are used for structural components, mechanisms, enclosures, etc. These metals increase the wear resistance and provide salvage for some components.

Laser surface treatment represents an alternative to the more conventional metal-based composite (MMC) manufacturing methods. In laser surface alloying, the powder mixture with a thin surface layer of the substrate is melted using a scanning laser beam. This leads to rapid solidification and surface layer formation [1]. The rapid cooling and solidification of the molten group produced by the laser can lead to the formation of various unbalanced phases and the fine-tuning of the microstructure. If the powder is injected into a molten solution contsaining non-melting particles, an MMC surface will form on the surface of the substrate [2]. The laser-treated hard coating has a lower surface energy than the reference, possibly due to the formation of hydrophobic carbide phases mainly on the sample surface. This also contributes to the improved anti-corrosion properties of this type of coating. It can be observed that the samples are not significantly oxidized during the LHT; otherwise, the surface energy would increase significantly [3]. Oxidation is often associated with the formation of poorly adhered phases, which flake or suffer delamination under very severe abrasive conditions. Although surface energy decreases when laser power is increased, the fraction of polarizing components increases, contributing to a better compatibility of the coating's surface with a wide range of oil-based paints and primers, as seen in various research papers on the wetting envelope process [4].

The surface is treated using laser irradiation, including laser hardening, alloying, and coating [5,6]. All these processes have in common the generation of specific thermal cycles in small, very localized areas of the part's surface, which then acquire new properties that allow it to resist wear, fatigue, and corrosion while retaining most of its other original properties [7].

Recent reviews of the principles and applications of laser processing refer to the use of lasers for hardening by acting as a controlled heat source [8]. The classical approach to modeling the heat flow caused by a dispersed heat source traveling over the surface

of a semi-infinite solid begins with the solution of a point source and integrates it with the beam surface. This method requires numerical procedures to evaluate it, such as the finite difference solutions of Shuja et al. [9]. The authors developed a 3D model of the heat flow, and various beam power and travel speeds were determined for the dimensional analysis of the heat flow during heat treatment and melting. However, the result is not easily distinguishable from complex shapes and also requires complex calculations.

Typically, the hard surface microstructure is refined by additional heat treatment for the purpose of stress relief or subsequent formation of hard phases (carbide, boride). Laser surface heat treatment is commonly used to treat carbon steel and alloy steel components due to its better directivity and time efficiency in reaching and maintaining the desired temperature.

By adjusting the surface hardness and wear resistance, the fatigue and corrosion resistance of the hard coatings can be improved [10,11]. The improvement in hardness after LHT can be attributed to the microstructural growth of austenite in systems that contain Fe-C, which is annealed in situ to martensite due to the high temperature gradient [12], thus avoiding the use of any liquid or gas. Then, LHT can promote the formation of carbides or borides in the microstructure of the material, which positively contributes to the increase in hardness and wear resistance. According to ISO-ASTM 52900 (2015) [13], AM is a process of joining materials layer by layer in order to create three-dimensional parts [14]. In recent years, AM applications have been expanded into several industry sectors. This is because technology offers opportunities to improve functionality, productivity, and competitiveness. In this regard, metallic AM has limitless potential, as it has recently been explored in the medical, aerospace, and automotive industries [15].

The effect of the post-processing technique on the microstructure and fracture strength of Inconel 718, fabricated using the AM process, was studied herein. Constructed samples contain gamma-column dendrites and small amounts of gamma + Laves eutectics in the interphase. After direct aging, which is a heat treatment process, heterogeneous gamma''/gamma' accumulates around Laves phases. After dissolution + aging, delta phases accumulate into Laves phases, and microdissociation decreases. After homogenization + dissolution + aging, the Lave phase almost disappears.

Products with complex shapes, flexibility in operation, and reduced production time can be produced by using the AM method. But they also face several challenges, such as poor surface condition, undesirable microstructural phases, porosity and defects, delamination, signs of wear, lack of hardness, reduced corrosion resistance, and reduced service life.

The objective of this paper is to correlate the microstructures and mechanical properties of the MC after laser heat treatment.

This paper also aims to extend the previous findings on laser treatment of various metallic materials by providing more detailed information on the effect of laser energy on the surface properties of the layer by open-arc deposition hard coating. In this regard, the variation of the microhardness and wear resistance of the surface is presented herein.

## 2. Research Methods

The ability to control and manipulate friction is important in many applications. From a practical point of view, it is desirable to be able to control the friction force so that the total frictional force is reduced (or improved). Therefore, the turbulent mode is eliminated and a smooth slide is obtained instead. Such control can be of great technological importance for micro-mechanical devices and computer drives, where the early stages of motion and stopping often exhibit unexpected slip or failure. Chaotic clubbing behavior may be desirable, e.g., on stringed instruments. Traditionally, the control of friction has been approached by chemical means, usually by supplementing the base lubricant with friction modifiers. However, the standard lubrication techniques used for large objects are expected to be less effective in the micro- and nano-worlds. Therefore, new methods of control and manipulation are required.

In this paper, MC was performed using a Luftarc 150 Ductil arc welding device by using four types of electrodes (Table 1) at a welding amperage of 700 A, a welding voltage of 40 V, and a welding speed of 40–45 mm/min. The width of the weld bead was 10 mm. The base metal was S275JR SR EN 10025-2: 2004. It was 20 mm thick. The coating was 5 mm thick for all samples. After coating, the samples were annealed at 600 °C in order to eliminate internal stress.

**Table 1.** Electrodes used for coating.

| Electrode | Chemical Composition [%] | | | | | | | | |
|---|---|---|---|---|---|---|---|---|---|
| | C | Si | Mn | Cr | Ni | Mo | W | P | V |
| El 62 H | 0.7–1 | 0.8–1.6 | 0.3–0.6 | 3.5–5 | 2.5–9.5 | 1.5–2.5 | 2–3 | 0.7–1 | - |
| E 6-60 | 0.8 | 0.4 | 0.5 | 2.8 | 2.2 | 1.2 | 3 | - | - |
| E 48 T | 0.7–1 | 0.8- | 0.3 | 3.5 | 2.5- | 1.5 | 2–3 | - | - |
| KD 31 | 0.7 | 1.6 | 0.6 | 5 | 6.2 | 2.5 | 2.3 | - | - |

The LHT was applied to four types of welding coatings, as presented in Table 1. Nine variants were used for the laser heat-treated samples with the Nd laser source: YAG-Rofin DY 570 Germany, manufactured by robot ABB, Sweden. Table 2 presents the intensity of the laser beam during the surface heat treatment. For the pumping of neodymium lasers, xenon or krypton lamps were employed. Krypton lamps are more efficient than xenon lamps for the pumping of neodymium lasers because they present a more powerful emission in the 810 nm absorption area of the neodymium. When the pumping energy exceeds 10 J, it is more efficient to use xenon lamps. The improvement in pumping efficiency is obtained through the use of alkaline additive lamps. For the flash-emitted radiation to produce light as efficiently as possible in the active environment, the assembly was mounted in a cylindrical reflector with a single or double elliptic section so that the lamp(s) and the active environment could be located at one of the focal points.

**Table 2.** Intensity of the laser beam and the sample notation.

| Intensity of Laser Beam [W] | 1400 | 1500 | 1600 | 1700 | 1875 | 2150 | 2425 | 2600 | 2700 |
|---|---|---|---|---|---|---|---|---|---|
| Electrodes Type | | | | | | | | | |
| El 62 H | [K-1] | [K-2] | [K-3] | [K-4] | [K-5] | [K-6] | [K-7] | [K-8] | [K-9] |
| El 6-60 | [V-1] | [V-2] | [V-3] | [V-4] | [V-5] | [V-6] | [V-7] | [V-8] | [V-9] |
| E 48 T | [N-1] | [N-2] | [N-3] | [N-4] | [N-5] | [N-6] | [N-7] | [N-8] | [N-9] |
| KD 31 | [00-1] | [00-2] | [00-3] | [00-4] | [00-5] | [00-6] | [00-7] | [00-8] | [00-9] |

The experimental research is focused on determining the influence of laser radiation on the structure and properties of welded, loaded layers and their opportunities for improvement. This influence was studied in two variants, as follows:

### 2.1. Hardening with Laser Beam

The thermal cycle of superficially hardening with a laser beam is very sharp, which means that the temperature in the layers varies very quickly when heating and cooling, and the heat treatment is without hold time.

The heating temperature for hardening is $0.6$–$0.7 \times T_{metling}$. In the short time of the action of the laser beam, the structure of the layers heated above $A_{C3}$ was transformed into austenite, having a very fine granulation. Because of the internal thermal stresses, this austenite would be hardened and transformed into inhomogenous martensite with a large number of dislocations. Because of chemical anisotropy, it is possible to keep a part of the residual austenite. Due to these reasons, after superficial LHT, it was possible for some internal stresses to appear. In this direction, austenite was transforming into martensite with internal tension and residual austenite. To avoid the appearance of cracks, it is very important to respect the optimal parameters. In the case of steel with high carbon

content and alloyed steel, it is very dangerous, and thus it is necessary to set the parameters very carefully.

*2.2. Surface Melting with Laser Beam*

In the case of surface melting, a very thin surface layer was melted, and after the laser beam was stopped, the solidification of the melted layer was very quick.

In the case of heating with a laser beam for surface melting, a large part of the energy was lost by reflection. To increase the energy efficiency, it was necessary to apply a film of absorbing materials of 30 μm thickness to the sample surface.

The YAG crystal has very good optical and mechanical features, with a very low laser effect limit and increased thermal conductivity. It is also able to function at a temperature of 20–25 °C in continuous or pulsating conditions with a high repeating frequency. The YAG crystals were obtained from a solution of melted salts in a closed platinum pot kept at a temperature of 1150 °C for 24 h and then cooled down to 750–850 °C at a rate of −4.3 °C per hour. The salt mixture composition was 3.4 mol% $Y_2O_3$, 7 mol% $Al_2O_3$, 4.15 mol% PbO, and 48.1% $PbF_2$.

The thermal properties of the YAG crystal with neodymium (pure) and with impurified neodymium are detailed in Table 2 for three temperature values.

The fluorescence transitions that may lead to laser emission are the ones that correspond to the transitions $^4F_{3/2} \rightarrow {}^4I_{9/2}$, $^4F_{3/2} \rightarrow {}^4I_{11/2}$, and $^4I_{3/2} \rightarrow {}^4I_{13/2}$. Their wave lengths were approx. 0.9, 1.06, and 1.35 μm, respectively [8].

The specific transition of the laser emission was located in a close infrared wavelength of λ = 1.06 μm. The width of this fluorescence line was greater than 30 A. It may be noticed that the Nd: YAG is a four-level laser whose advantage is that it facilitates a high amplifying coefficient. Also, given the fluctuation threshold, the Nd: YAG laser may function in a continuous condition with the aid of plain water circulation-based cooling.

An intense radiation of white light populated all levels located above $^4F_{3/2}$, from where the ions turned back towards it through non-radiating transitions. The $^4I_{11/2}$ inferior level was located at $2 \times 10^3$ $cm^{-1}$ above the fundamental level $^4I_{9/2}$ and was void at room temperature, which facilitated the achievement of the pumping threshold. In addition, the energy levels of the neodymium ion were less sensitive to the features of the imperfections of the crystalline network. This is because they were involved in the changes in the electronic configuration of the internal layers instead of the electrons of the external layers, as in the case of $Cr^{3+}$. This insensitivity of the $Nd^{3+}$ ion related to what is in its environment facilitates the choice of yttrium-aluminum granite as a matrix, as well as its insertion into an amorphous matrix such as glass, for instance [16].

The life span of the neodymium ion's fluorescence depends both on the quantity and quality of the doping as well as on the composition of the host material.

The efficiency of the Nd: YAG laser is approx. 4%. It may emit, in certain conditions, a power on the order of 5 kW. In continuous conditions that cater to laboratory needs, powers of 20–100 W are exceeded. The Nd: YAG laser may function in an open condition or in a mode coupling condition. In the pulsating condition, the average power of the Nd: YAG generators is approx. equal to the output power in continuous conditions. The efficiency of this type of laser may be increased if, in the YAG crystal, there are also introduced $Cr^{3+}$ impurities that substitute the $Al^{3+}$ ion. In this case, a non-radiating energy transfer occurs from a $Cr^{3+}$ ion pumped on the $^2E$ level to a $Nd^{3+}$ ion that develops into the $^4F_{3/2}$ state.

Usually, a Nd: YAG laser consists of a neodymium-doped cylinder bar with a 5–20 mm diameter and 50–250 mm length, which represents the active optical environment. The bar heads are optically labored and have reflecting layers that form the resonant cavity of the laser. The optical pumping may be performed with a coherent source by using the emission of another laser or with the aid of a "classic" lamp with continuous emission or impulses (flash) that emit light on a wide scale in all directions of the environment.

The laser radiation presents a series of unique properties, some of which play an extremely important role in the utilization of the laser beam for thermal treatments. The

laser radiation is practically monochromatic, meaning that it has a narrow spectrum range and may be replaced by a certain wavelength and frequency.

Consequently, the power spectrum density of the laser radiation exceeds, by a few lengths, the power density of radiation from other known electromagnetic energy sources with extremely wide spectrum ranges [17].

The basic features of the laser beam, such as polarizing, glowing, and time distribution of the beam, are grouped into those most characteristic for thermal treatments, namely, the capacity to concentrate a large amount of energy into an extremely short time span on a given surface and in a small substance mass.

## 3. Results and Discussion

The obtained results were evaluated with a PMT 3 microhardness tester, an electronic microscope (Nova Nano SEM), and a chemical analyzer (EDAX Orbis Micro-XRF).

The laser-treated layer thickness was 0.2–0.9 mm, and it depended on the intensity of the laser beam. The structures obtained using the SEM technique are presented in Figures 1–4.

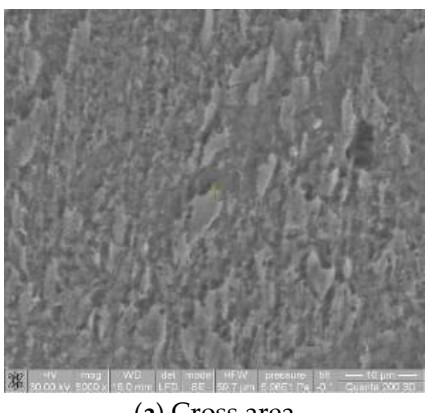

(**a**) Cross area

(**b**) Coating layer

**Figure 1.** SEM structures of MCs with El62H + LHT 1600W.

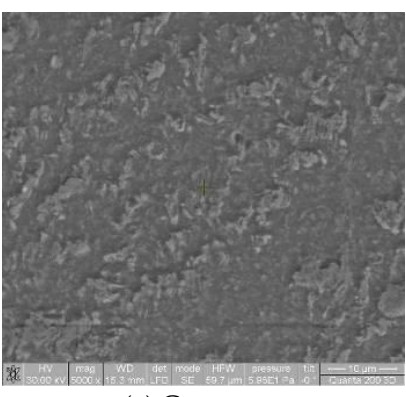

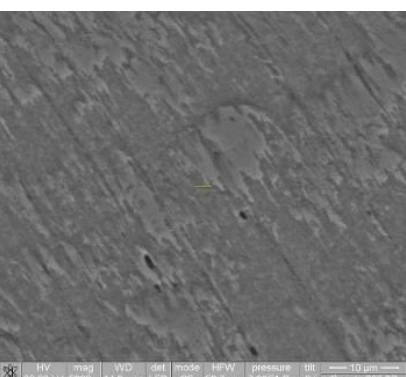

(**a**) Cross area

(**b**) Coating layer

**Figure 2.** SEM structures of MCs with El 6-60 + LHT 1700 W.

The increase in Cr concentration determines the formation of the chemical compound Cr 2 in the ferrite–pearlite matrix, especially in the case of electrodes 6–60 and KD31. In the case of the MC with the 48T electrode, which has a low Cr concentration, no chemical compound formation could be observed. Figure 5 presents the chemical composition obtained with the EDAX chemical analysis. By considering the microstructure of the test coated with E 62H, which includes a high concentration of C, it was possible to observe the ferrite–pearlite structure of the base fabric and the structure of the needle-shaped martensitic structure. In the case of the other five terminal types, for the one with more C,

more pearlite structures were obtained. The base fabric includes a ferrite–pearlite structure with lamellar pearlite.

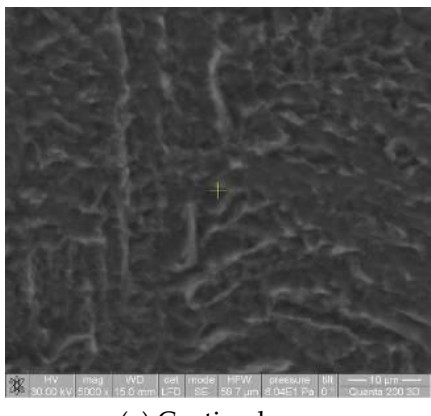

(**a**) Coating layer

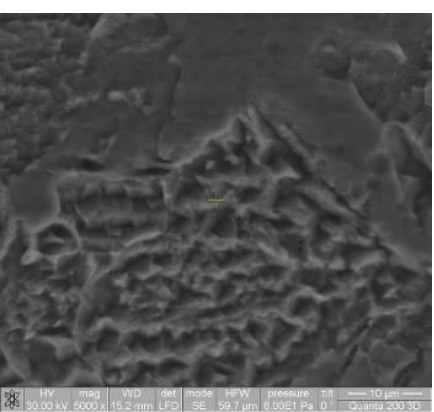

(**b**) Coating layer

**Figure 3.** SEM structures of MCs with El48T + LHT 2700 W.

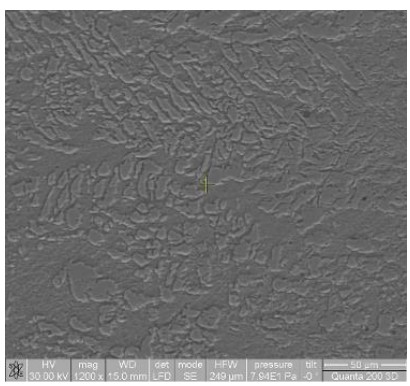

(**a**) Cross area

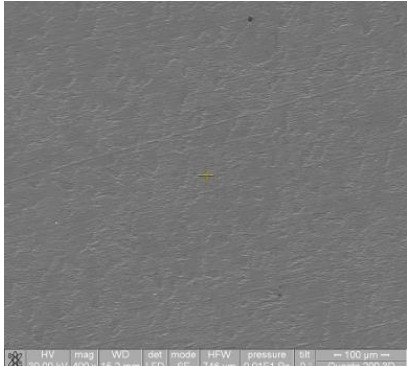

(**b**) Coating layer

**Figure 4.** SEM structures of MCs with El KD31 + LHT 2425 W.

The furthest layer was characterized as the compound locale, and the inward layer underneath as the dissemination locale. Each zone contributed to a moved-forward execution by progressing particular specialized properties, specifically wear resistance, grease resistance, erosion resistance, and weakness resistance. By considering the microstructure of the test coated with E 62H, which included a high concentration of C, it was possible to observe the ferrite–pearlite structure of the base fabric and the structure of the needle-shaped martensitic structure. In the case of the other five terminal types, for the one with more C, more pearlite structures were obtained. The base fabric included a ferrite–pearlite structure with lamellar pearlite. Based on the range of appearance, columnar dendrite structures were observed within the Fe network with a good boundary. From these structures, the following execution benefits were realized: exceptional running properties, anti-wear properties, and decreased propensity to terrible erosion. By examining the SEM structures, as seen in Figures 1–4, microhardness, as seen in Figure 6, and scraped area resistance, as seen in Figures 7–10, a strong bond could be observed, especially between microhardness and wear resistance. After a more escalated laser surface warm treatment, the structure of the fabric was modified, coming about in halfway change of ferrite into pearls with a better C concentration, which also leads to better mechanical properties. In the case of highly concentrated laser surface warm treatment of more than 2 kW, the arrangement of remelting structures could be observed, resulting in altogether expanded mechanical properties [18–20].

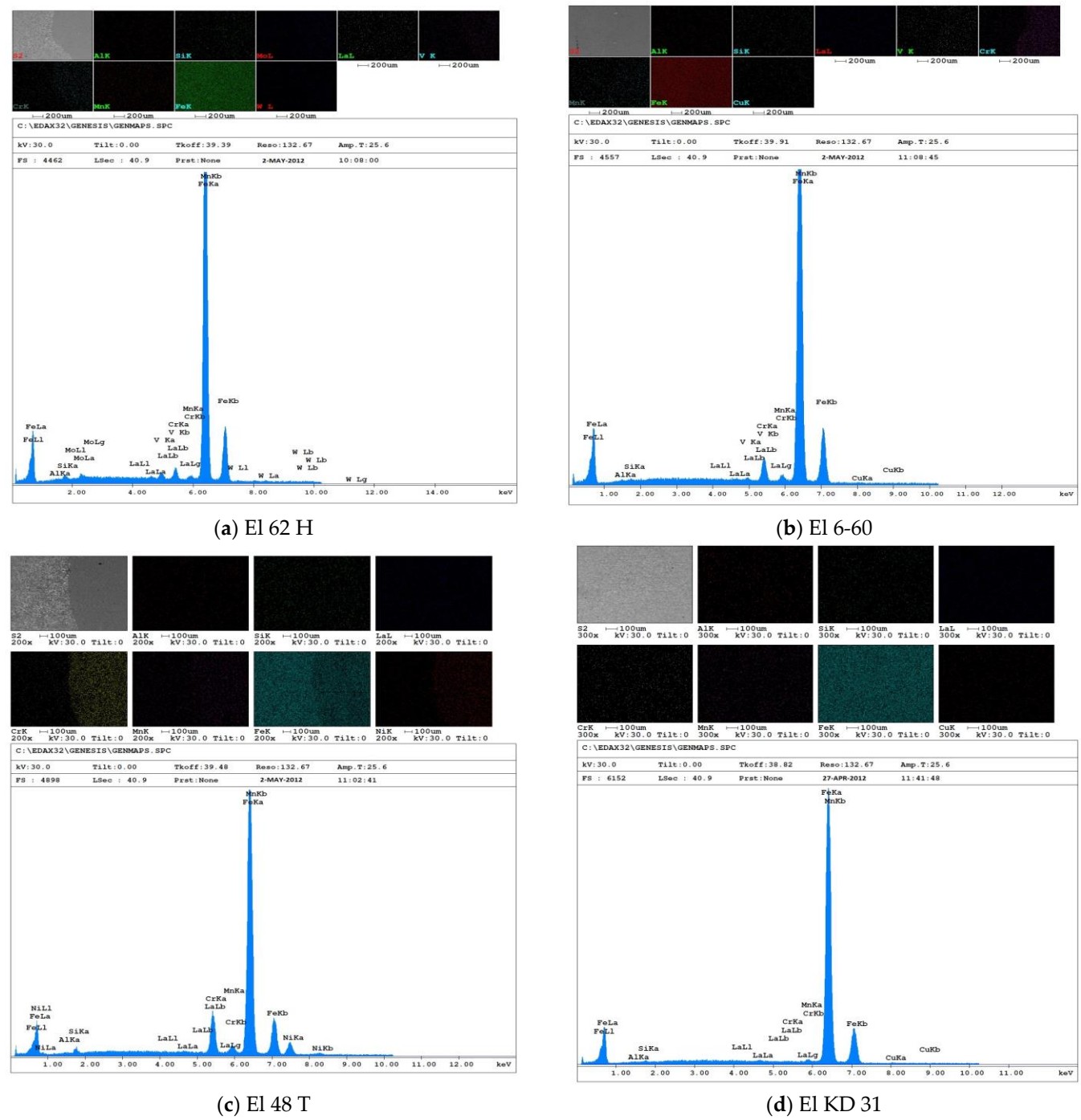

(**a**) El 62 H

(**b**) El 6-60

(**c**) El 48 T

(**d**) El KD 31

**Figure 5.** Chemical composition of MC layers.

Evaluation of the mechanical properties was performed for microhardness and wear resistance. The obtained microhardness is presented in Figure 6.

In these conditions, the natural physical properties of the compound, specifically hardness and lubricity, altogether move forward the sliding and running behavior and, in this way, increment the cement's wear resistance. The stage composition of the mixed locale that presented the finest wear resistance was mainly composed of an epsilon nitride stage (a single stage is preferable), which also consisted, in a small part, of the gamma essential stage. It was shown herein that the scraped area resistance depends on the relative bundles of the rough and composite locales [21].

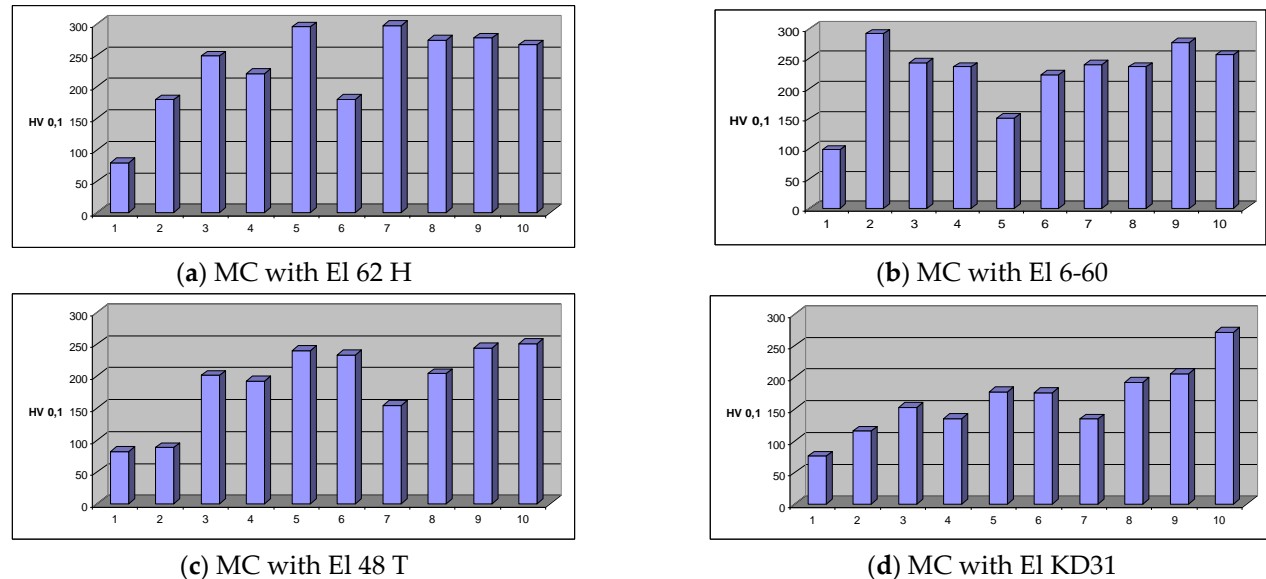

(**a**) MC with El 62 H

(**b**) MC with El 6-60

(**c**) MC with El 48 T

(**d**) MC with El KD31

**Figure 6.** Microhardness of the MC layers.

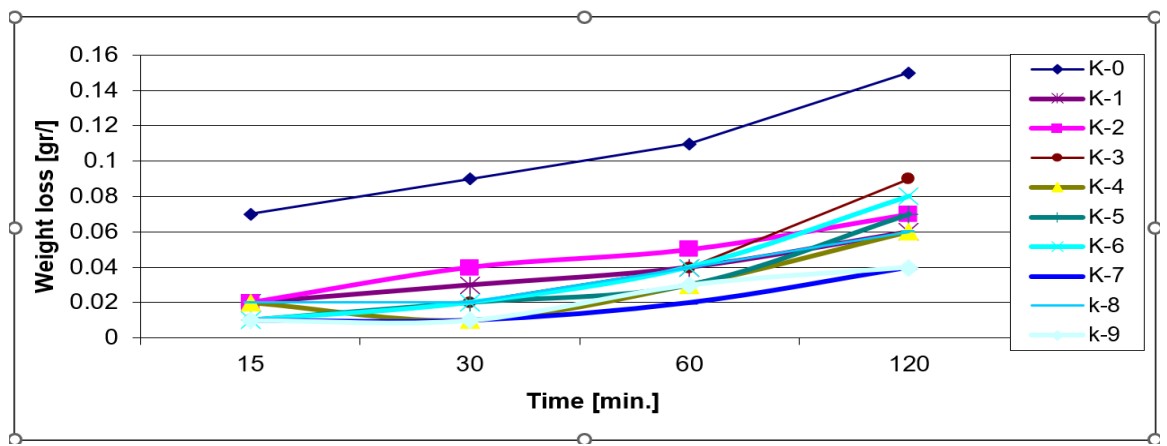

**Figure 7.** Weight loss of MC layers of MC with El 62 H.

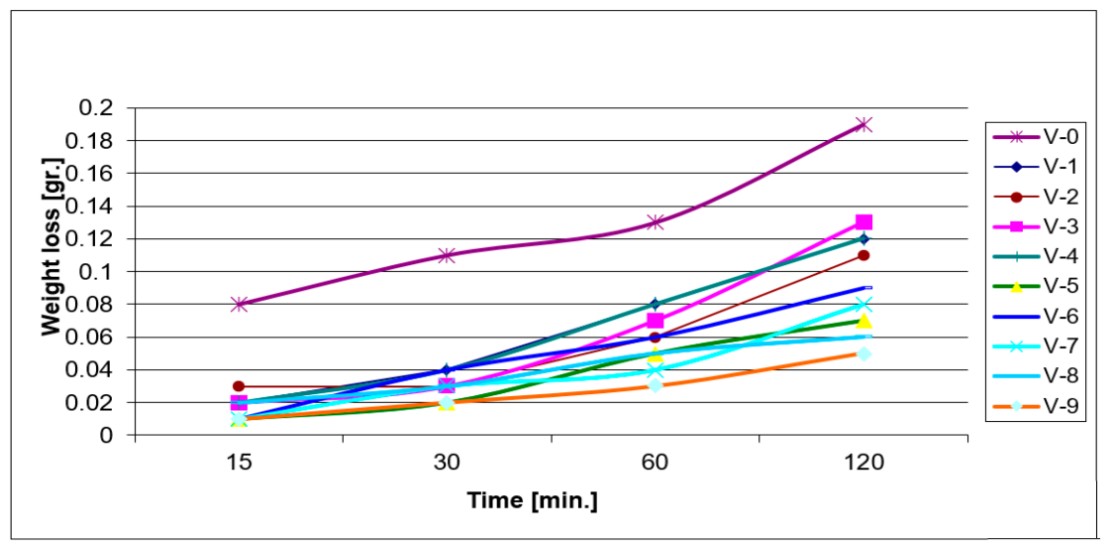

**Figure 8.** Weight loss of MC layers of Mc with El 6-60.

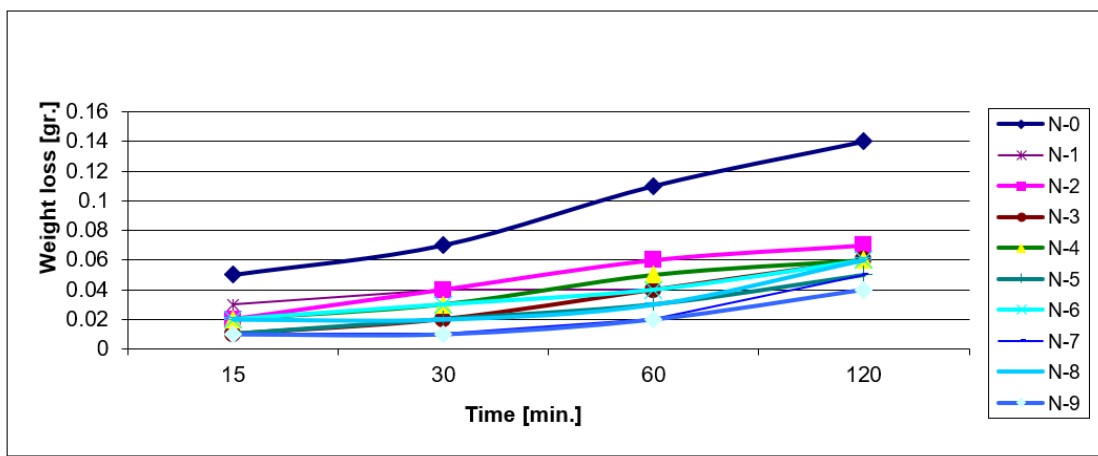

**Figure 9.** Weight loss of MC layers. MC was performed with E 48 T.

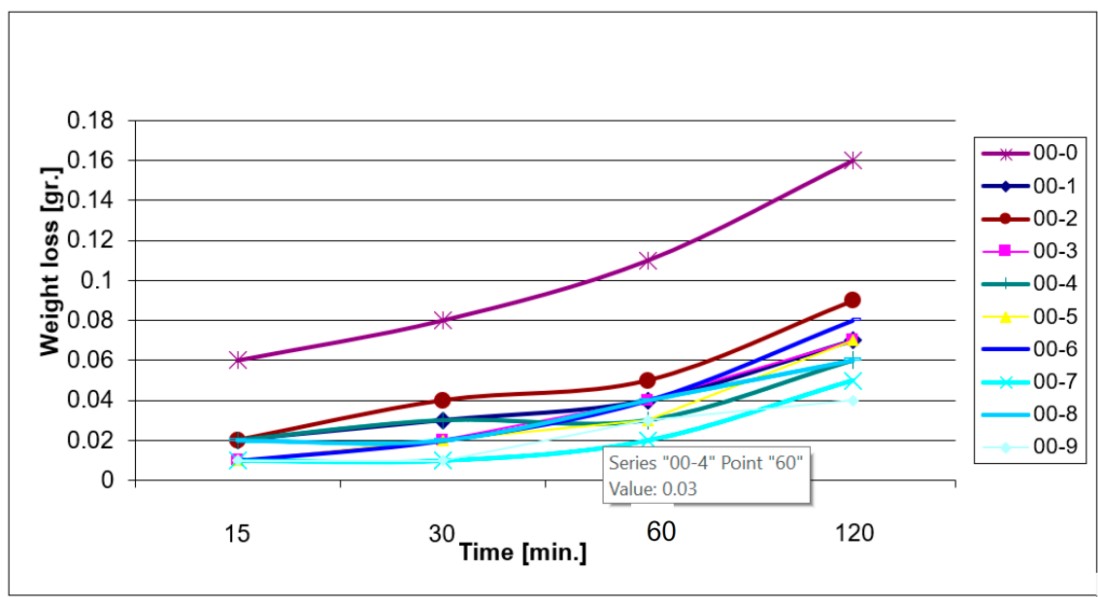

**Figure 10.** Weight loss of MC layers. MC was performed with EL KD 31.

Abrasion resistance was tested after 15, 30, 60, and 120 min of abrasion and sample weighing. The loss of mass was determined using an "Oertling-England" balance with an accuracy of 102 g. The results are shown in the abrasion graphs in Figures 7–10.

By analyzing the wear resistance of the samples, the laser-treated hard facings generally present 30%–57% lower wear rates than the reference untreated samples (K-0 and N-0, respectively). This could be caused by the formation of hard phases (chromium carbides, martensite, and residual austenite) in the bulk of the cladding [22].

Scraped area resistance is possibly the foremost critical property determined by the surface treatment. The wear resistance of the composite zone depends on the type of anti-wear framework, especially if it is cement or grating wear. Colloidal wear happens when two components are in relative motion in a non-abrasive base medium [23].

The obtained coating surfaces exhibited significant roughness values, as demonstrated by 3D AFM topographic images [9]. An increase in the surface roughness profile was observed when increasing laser power, possibly due to the constitutive morphological reconstruction, aided by the improved elemental scattering. In addition, the presence of intermetallic compounds or carbides on the surface of the sample, embedded in the ferrite matrix, could also be noted. Surface energy is considered an important parameter that is directly related to the corrosion resistance of the sample and the ability of the

surface to better interact with different species with similar surface energy values (for example, primers, paints, other metals, polymers, etc.). For the reference samples, the main component of the surface energy is the dispersion component. It can be noted that the K-0 sample had the lowest surface energy, possibly due to the higher amount of chromium present on the surface of the material.

## 4. Conclusions

In this paper, it was shown that characterization of hard surfaces obtained after coating with an El 62 H electrode, mainly in solid solution enhanced with brown shades (ferrite, austenitic, martensite) and intermetallic compounds (possibly Fe, Mn3Si), takes place at the grain boundary. Most of these compounds were observed to be present in the treated sample, emerging with the highest laser energy densities. It was noted that, in the case of coatings with low Cr content, the heat treatment of the laser beam affected the precipitation of C and Cr carbides, increasing with the intensity of the laser beam. In addition, the formation of needle-like martensitic structures was observed [24].

The microstructure of the hard surface coating obtained with the E 48 T electrode has revealed the presence of carbides (spherical at N-0, arranged at grain boundaries of N-6 and N-7). The largest amount of carbide was found in the N-7 sample, which presented the highest laser power density. The EDS elemental mapping of the sample cross-sections also supports the formation of chromium carbide during the LHT in the case of a hard coating deposited by the El 62 H electrode, with the highest chromium content. These carbides typically have a higher density relative to the rest of the coating material, which determines their migration into most of the coating. This is also aided by a high chromium concentration gradient between the coating and the base material. The reason why the apparent chromium content at the cross-sectional surface decreased with increasing laser power could be due to the fact that no significant diffusion of this element was found in the base material. The surface was richer in Si and Mn, making the material more resistant to corrosion (the formation of silicide and intermetallic compounds), while the carbide ensured a higher wear resistance of the hard coating. For the hard coating produced by the E 48 T electrode, a low chromium content (2.5% $w/w$) determined a less pronounced amount of carbide formation compared to that of the L-types. The presence of Cr, W, V, and Mo on the surface of L-type test pieces can also contribute to their increased resistance to corrosion. Moreover, the sharp, clear separation boundary between the substrate material and the hard coating can act as a diffusion barrier, thereby retaining the Cr, W, Mo, V, and Mn on the sample surface.

The formation of hard phases in most laser-treated hard coatings can be explained by the high values of microhardness, as shown in Figure 1a,b. The formation of martensite, as well as chromium and carbide mixtures, can be explained by the remarkably high microhardness value in samples K-6 and K-7, as seen in Figure 2b, which is identical to that of pure martensite.

**Author Contributions:** Conceptualization, A.O. and T.M.-P.; methodology, A.O.; software, P.V.; validation, A.O., T.M.-P. and P.V.; formal analysis, A.O.; investigation, A.O.; resources, A.O.; data curation, A.O.; writing—original draft preparation, A.O.; writing—review and editing, T.M.-P.; visualization, P.V.; supervision, T.M.-P.; project administration, A.O.; funding acquisition, T.M.-P. All authors have read and agreed to the published version of the manuscript.

**Funding:** This paper is supported by Sectorial Operational Programme Human Resources Development (SOP HRD), finance from the European Social Fund and by Romanian Government under project number POSDRU/89/1.5/S/59323.

**Institutional Review Board Statement:** Not applicable.

**Informed Consent Statement:** Not applicable.

**Data Availability Statement:** Not applicable.

**Conflicts of Interest:** The authors declare no conflict of interest.

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
