# Peer review of "Wearing Resistance of Metal Coating Layers after Laser Beam Heat Treatment"

_coatings, doi:10.3390/coatings13091645_

Round 1
Reviewer 1 Report (Previous Reviewer 2)
In this paper, the effect of laser heat treatment on the wear resistance of coatings is discussed. However, the following problems need to be solved before publication:
1. Abstract: The most important results should be present in the abstract.
2. It is recommended that all tables in the paper use three-line tables with centered headers.
3. Electrode types in Figures 1-5 do not correspond to Table 2. For example, Table 2 is E 6-60 and Figure 2 is El 6-60.
4. Page 7, lines 230. Punctuation is used incorrectly, please check carefully.
5. Page 8, lines 242. What does “an accuracy of 10-2 grams” mean? Is that supposed to mean “an accuracy of 10-2 grams”?
6. It is essential to clarify the names of the electrodes. For example, is it E 48 T (page 10, lines 287) or E48 T (page 10, lines 301)?
7. Conclusion: This paper does not provide a point-by-point account of the conclusions, so if possible please refine the conclusions section and break it down into points. Moreover, it is necessary to make a general summary at the beginning of the conclusion.
The author is requested to carefully edit the English language of the full text.
Author Response
Dear Editor and Reviewers,
Thank You for Your observation.
Article was revised included abstract.
The paper including description of used laser on experimental research.
Mentioned observation was resolved.
The paper was restructured, including conclusion.
English Language was improved.
On behalf authors,
Best regards,
Assoc.prof.dr. Olah Arthur

Reviewer 2 Report (New Reviewer)
In this paper, the authors carried out a controlled experiments in understanding the impact of laser intensity and electrode chemical composition on metal coating layers, which is characterized with SEM/EDX, microhardness and wear resistance. It also attempts to link the micro-structures observed from SEM to the mechanical properties. The experimentation and characterization are sound. However, for publication, improvements are needed on the presentation of data, and the explanation on the observation in the discussion section.
1) In the Introduction portion, some more reference and introduction on previous studies on Laser treatment on coating layer, and particularly the characterization of the microstructure using SEM, AFM, etc.. Also, some comments on how this study is novel and differentiates from existing are needed.
2) In the Results and discussion section, Fig. 1-4 can be combined for better presentation.
3) Captions of Fig. 1-4 quote and compare conditions with different electrode compositions as listed in Table. 1, and different laser intensities. Why different laser intensities are selected for comparison? These four figures also lack proper description to link the microstructures to mechanical properties presented in later figures and materials. Discussion and conclusions around them (line 216-227) are hard to follow.
4) More explanation is needed on the plots in Figure 6, and particularly how it relates to the SEM/EDX images.
5) In Figure 7-10, legends are not well-formatted and have missing entries. Y-axis on Figure 7 may have a typo on its unit.
6) Description of the Laser source and its operation, as well as other instruments (line 133-204) may be reorganized into Research methods section
1) From line 78 to line 79, two "On the other hand" are used at the beginning for two consecutive sentences. Same issue for line 93 and 95.
2) Line 85, "moreover" and "also" are redundant.
3) Line 121, based on the content, "transforming in" should be "transforming into"
4) On Line 220: "it possible to observe..." seems missing an "is"
5) Line 271 "... is considered an..." should be "... is considered as an..."
Author Response
Dear Editor and Reviewers,
Thank You for Your observation.
Article was revised included abstract.
The paper including description of used laser on experimental research.
Mentioned observation was resolved.
The paper was restructured, including conclusion.
English Language was improved.
On behalf authors,
Best regards,
Assoc.prof.dr. Olah Arthur

Reviewer 3 Report (New Reviewer)
1. A brief introduction to background should be added to the abstract.
2. Page 3 line 114, delete “of”.
3. Page 4 line 169, “catter to” should be “cater for” instead.
4. Figure 1 through figure 4 should be combined and named as figure 1 (a-h), as they are all SEM images.
5. It is hard to read texts in Figure 5, would be better to present the composition in a table.
6. Figure 6, x and y axis should be clearly labeled.
7. Page 8 line 242, “10-2” should be “10-2”
8. Figure 7, 8, 9 and 10: there are many curves in the figures without any label.
9. The second paragraph in the conclusion should be moved to the discussion part.
10. How are material phases determined from EDS results? For example, page 10 line 290, “The EDS elemental mapping of the sample cross-sections also supports the formation of chromium carbide during the laser heat treatment”. How is chromium carbide determined from EDS results? Could it be chromium and carbon instead?
Author Response
Dear Editor and Reviewers,
Thank You for Your observation.
Article was revised included abstract.
The paper including description of used laser on experimental research.
Mentioned observation was resolved.
The paper was restructured, including conclusion.
English Language was improved.
On behalf authors,
Best regards,
Assoc.prof.dr. Olah Arthur

Reviewer 4 Report (New Reviewer)
The authors have undertaken a study of the influence of laser treatment on welding beads deposited on a carbon steel plate. In my opinion, the work is very deficient and does not rigorously address the aspects of a quality research work on this subject.
1. The introduction is scattered and does not provide any relevant information about the research carried out. Are there no similar experiences in the literature? The procedure to generate a weld overlay has not been justified and other techniques and possibilities to achieve similar objectives have not been described. The reason for the selection of this type of electrode is not known. The introduction is not adequately contextualised with the tests carried out in the work.
2. A material and methods section should be included describing all the details of the equipment and materials, for example: (i) type of laser used, including its wavelength, power, pulse duration (if applicable) and any other relevant characteristics, (ii) the total energy used during treatment and the fluence (energy per unit area) applied on the surface of the material should be described, (iii) details of the preparation of the samples prior to treatment, (iv) the experimental setup should be explained, this includes the geometry of the sample, (iv) include treatment parameters such as scanning speed, number of passes, inter-spot distance, angle of incidence, substrate temperature, etc., (v) describe the technique used to make the weld beads, including electrode movement, welding sequence, type of joint (overlap, fillet, etc.), and any other relevant aspects, and any other relevant aspects. ), and any other relevant aspects, (vi) indicate how the quality of the weld beads was assured, such as visual inspection, non-destructive testing, penetration testing, etc. (vii) equipment used to obtain SEM photographs, micro-hardness, abrasion equipment.
3. I do not see the point in describing in such detail the laser formation used as this is not the objective of the work and even less so in the chapter on results and discussion. The unit is kW and not KW (line 169).
4. Table 2 should be in the material and methods chapter. The images in figure 2 should include a scale bar. The images add little or nothing of interest. It would have made sense to carry out a metallographic analysis, it is highly doubtful whether the metallographic structure of the steel can be determined by SEM.
5. The images in figure 5 are unreadable and must be modified to be interpreted.
6. The results in figure 6 do not include error bars.
7. The figures on coating wear are confusing and there are a greater number of curves than indicated in each figure.
Author Response
Dear Editor and Reviewers,
Thank You for Your observation.
Article was revised included abstract.
The paper including description of used laser on experimental research.
Mentioned observation was resolved.
The paper was restructured, including conclusion.
English Language was improved.
On behalf authors,
Best regards,
Assoc.prof.dr. Olah Arthur

Round 2
Reviewer 1 Report (Previous Reviewer 2)
1. Tables in the paper are suggested to adopt the format of three-line table.
2. Resolutions of Figure 2 and Figure 3 are too low, figures with higher resolutions are strongly recommended.
Minor editing of English language required
Author Response
Thank You for Your observation.
On behalf authors,
Best regards,
Assoc.prof.dr. Olah Arthur

Reviewer 4 Report (New Reviewer)
The authors have undertaken work to study the influence of laser treatment on weld beads deposited on carbon steel plate.
The authors still do not answer most of the questions raised. The chapter entitled Research Methods does not clearly show the form of the experiments and with the information given they could not be replicated by another group of researchers. This is a basic pillar of any research work. Furthermore, it includes sections that contribute very little to the proposed research and should be eliminated or synthesised such as "Surface melting with laser beam" as already indicated in the previous review.
The authors have not taken the precaution of marking new paragraphs or parts with a different colour. Please ensure that this is done for subsequent versions.
Authors are requested to respond specifically to the following questions:
1. It is not known where in the text the authors have provided new comments on the rationale for generating a weld contribution coating, nor are there any comments on the rationale for the selection of the electrode type used in the research.
2. It is strongly recommended to include a section on materials and methods with a new or reorganised Research Methods section describing all the details of the equipment and materials for the research and providing a clearer and more organised account of the work carried out. Again, we propose that the following information be included, where this has not already been done, and reorganised:
(i) type of laser used, including its wavelength, power, pulse duration (if applicable), laser spot dimensions and any other relevant characteristics.
(ii) total energy used during treatment and the fluence (energy per unit area) applied to the surface of the material.
(iii) sample preparation prior to treatment.
(iv) experimental set-up, this includes sample geometry, fixturing, position of the laser equipment, etc.
(v) parameters of the laser treatment such as scanning speed, number of passes, distance between points, angle of incidence, substrate temperature, etc.
(vi) technique used to carry out the weld overlay, including electrode movement, welding sequence, type of joint (overlapping, filleting, etc.), and any other relevant aspects,
(vii) quality of weld coatings, e.g. by visual inspection, non-destructive testing, penetration testing, etc.
(viii) equipment and methods used for all tests used.
3. The results in figure 6 do not include error bars.
Author Response
Thank You for Your observation.
On behalf authors,
Best regards,
Assoc.prof.dr. Olah Arthur

Round 3
Reviewer 4 Report (New Reviewer)
May be published
This manuscript is a resubmission of an earlier submission. The following is a list of the peer review reports and author responses from that submission.
Round 1
Reviewer 1 Report
There are some significant defects in this manuscript. The title is too vague and lacks focus. The abstract is too rough, and it is difficult to understand the main points of the manuscript through it. There is a lot of redundant information in the manuscript, such as description of lasers. The overall manuscript is not rigorous, and lacks enough analysis and discussion of experimental results. The conclusion needs to be condensed.
Quality of English Language is needed to be improved
Author Response
Thank You for Your observation. Article was revised included abstract. The paper including description af used laser. The paper was restructured, and impruved English Lenguage.
Reviewer 2 Report
In this paper, the effect of laser heat treatment on the wear resistance of coatings. However, the following problems need to be solved before publication:
1. It is recommended that all tables in the paper use three-line tables with centered headers..
2. Page 2, lines 90. Check the spelling of “e.g. g.” in “On the other hand, chaotic clubbing behavior may be desirable, e.g. g., in stringed instruments.”
3. Page 2, lines 95. Whether “Metal coating was performed using a Luftarc 150 Ductil arc welding device, using four types of electrodes (Table 1).” needs to be indented.
4. The resolutions of Figures 1 to 4 are too low, and it is recommended to use SEM micrographs with higher resolution.
5. Page 7, lines 209. “The results are shown in the abrasion graph, figure 7…10” should be replaced by “The results are shown in the abrasion graph (figure 7-10).”
6. The figure annotation given in Figure 7-10 is incomplete and it is recommended that it be supplemented.
7. Conclusion: This paper does not provide a point-by-point account of the conclusions, so if possible please refine the conclusions section and break it down into points. Moreover, it is necessary to make a general summary at the beginning of the conclusion.
8. Taken overall, the paper suffers from a lack of structure. In addition, there are grammatical and spelling errors throughout the text, the quality of English needs improvement.
The quality of English needs improvement.
Author Response
Thank You for Your observation. All Your observation wae revised included abstract. was restructured of colculion and impruved English Lenguage.
Reviewer 3 Report
This paper appears to describe effect of laser heat treatment on metal surface modification. Unfortunately, purpose of this work and evaluation of results including comparison with other related studies are not well discussed here, and novelty of this work cannot be evaluated. This paper cannot be recommended for publication in Coating.
Author Response

(The authors gave the same response as above.)
